# Training Using a Commercial Immersive Virtual Reality System on Hand–Eye Coordination and Reaction Time in Young Musicians: A Pilot Study

**DOI:** 10.3390/ijerph18031297

**Published:** 2021-02-01

**Authors:** Sebastian Rutkowski, Mateusz Adamczyk, Agnieszka Pastuła, Edyta Gos, Carlos Luque-Moreno, Anna Rutkowska

**Affiliations:** 1Faculty of Physical Education and Physiotherapy, Opole University of Technology, 45-758 Opole, Poland; a.rutkowska@po.edu.pl; 2Descartes’ Error Student Research Association, Faculty of Physical Education and Physiotherapy, Opole University of Technology, 45-758 Opole, Poland; maadamczyk1998@wp.pl (M.A.); agnieszka.pastula037@gmail.com (A.P.); edytag565@wp.pl (E.G.); 3Nursing and Physiotherapy Department, University of Cadiz, 11003 Cadiz, Spain; carlos.luque@gm.uca.es

**Keywords:** VR, virtual reality, TMT, hand–eye coordination, reaction time, immersion

## Abstract

The implementation of virtual reality (VR) opens up a wide range of possibilities for the development of dexterity, speed and precision of movements. The aim of this study was to investigate whether immersive VR training affected the hand–eye coordination and reaction time in students of the state music school. This study implemented a single-group pre-post study design. This study enrolled 14 individuals, submitted to a 15 min training session of the immersive music game “Beat Saber”, once a day for 5 consecutive days. The plate-tapping test (PTT) and the ruler-drop test (Ditrich’s test) were used to assess the reaction time. Trial-making test (TMT) A and TMT B were used to assess coordination and visual attention. Analysis of the results showed a statistically significant improvement in hand–eye coordination and reaction time of music school students using the TMT-A (*p* < 0.002), TMT-B (*p* < 0.001), Ditrich’s test for the non-dominant hand (0.025) and PTT (0.0001) after applying a week-long training period in immersive VR. The results obtained in the present study show that the VR system, along with the immersive music game, has the potential to improve hand–eye coordination and reaction time in young musicians, which may lead to the faster mastering of a musical instrument.

## 1. Introduction

Training for better hand–eye coordination is essential in the modern age. Everything from driving to typing requires dexterity and improving it only helps us in other avenues. Interactions with objects and people play a crucial role in understanding how the brain creates internal models of the environment and generates movement within the space. Hand–eye coordination can be defined as the ability to perform activities that require the simultaneous use of the eyes and hands [1]. This is a complex cognitive proficiency, since it requires the hands to be guided according to the stimuli the eyes receive. Movements of the hands and eyes in everyday life occur in different contexts. Hand movements can be made toward a visual target shortly after its presentation or after a longer delay; alternatively, they can be made to a memorable destination [2]. The visual–motor reaction times of the hands reflect the integration of visual information, perceptual decisions and motor movements in order to perform a specific task [3]. The factors that affect the reaction to the stimulus include age, gender, distractions, personality or drugs [4]. Time of reaction is considered to be a reliable indicator of the speed and efficiency of mental processes and is an important variable in the behavioral sciences [5]. Despite this great interest, many problems arise with the implementation of response time, especially in differential and developmental research [6].

Music is one of the fields of multidisciplinary education. It has been shown that music is a powerful stimulator of the brain. Besides auditory cortices, motor regions, such as the supplementary motor area and the cerebellum, are also involved during musical activities, including both playing and listening [7]. Musical activity influences physical development in which skills such as coordinating movements, tempo, rhythm, balance and precision are essential. Music classes contribute to the development of movement by fostering better space orientation and the skill of harmonious, conscious and purposeful movement [8]. Musical movement activity improves the coordination of the muscle and central nervous system, which leads to better physical and mental functioning [9]. Musical-motor classes increase the level of auditory–motor coordination and the level of selected motor skills, as well as reinforce the functional coupling of these regions, which has been evidenced by many neuroimaging studies [10,11]. The training and practice of musicians require the simultaneous integration of multimodal sensory and motor information in the sensory and cognitive domains, combining skills in auditory perception, kinesthetic control, visual perception and pattern recognition [7,12]. Classes in rhythm and musicianship complement the education of young people, develop an immediate response to music and sensitize and build musical awareness [13].

With the development of technology, new opportunities have emerged to improve hand–eye coordination and reaction time using technical novelties, such as virtual reality (VR). Virtual reality (VR) has been defined as the “use of interactive simulations created with computer hardware and software to present users with opportunities to engage in environments that appear and feel similar to real-world objects and events” [14]. VR has been used in most educational environments as an opportunity to support many learners [15]. VR enables the user to experience immersion in a virtual world, thus being able to follow the world generated for their own use and experience things that are not available in real life. The benefits of transferring to the virtual world through the use of immersion provide the opportunity to target the training using the attractiveness of the imaginary environment [16,17]. As a result, users become more involved and motivated to train. In particular, the tactical immersion characteristic of arcade games, which requires attention and dexterity in order to make some moves correctly, can be used in the training process [18].

The analysis of the literature on the use of VR for hand–eye coordination has shown a small number of papers describing the use of such a type of training for physicians in surgery simulators [19,20]. Thus, we found a scarcity of literature evaluating how immersive VR training affected the hand–eye coordination in other professions requiring high coordination. On the basis of our own experience and literature review, we have chosen a group of young musicians to evaluate the effectiveness of training in immersive VR for hand–eye coordination and reaction time. Our aim was to investigate whether immersive VR training has the potential to improve hand–eye coordination and reaction time and to analyze the strengths and weaknesses of this system. We hypothesize that the implementation of VR opens up a wide range of possibilities for the development of dexterity, speed and precision of movements, which are crucial for mastering a musical instrument. The second objective was to assess the energy expenditure of such training. We hypothesize that VR can be an excellent platform, especially with rhythm games, that taps into the part of our brain responsible for hand–eye coordination and reaction time.

## 2. Materials and Methods

### 2.1. Participants

This study implemented a single-group pre-post study design. This study enrolled 14 individuals aged 14 to 19 years (8 females, 6 males, age = 17.6 ± 1.1 years, body mass index = 22 ± 2.6). All participants signed a written informed consent prior to their participation in the study. The study included students of the State Music School of I and II degree in Opole, who declared as their leading instrument: piano—nine persons; violin, cello and percussion—three persons; trumpet, straight flute, trombone, double bass, clarinet and organ—two persons. In all, 43% of respondents used more than one instrument. The inclusion criteria included individuals aged 14 to 20 years and students of the music school of second degree. The exclusion criteria included diagnosed neurological diseases, fear of restriction of vision or putting on goggles, diagnosed musculoskeletal diseases or injuries to the locomotive system. The study adhered to the Declaration of Helsinki [21], ethical approval was obtained from Bioethics Committee of the Opole Chamber of Physicians on the basis of Resolution No. 243 of 6 April 2017 and the study was registered in ClinicalTrials.gov (NCT04662983).

### 2.2. Intervention

The participants were submitted to a 15 min training session of the immersive music game “Beat Saber” (Beat Games, 2019), once a day for 5 consecutive days at the same time. The research station consisted of HTC Vive Pro (HTC Corporation, New Taipei, Taiwan) headset along with a connected laptop. The device enables high resolution and high fluidity (90 Hz) images to be displayed. The organic light-emitting diode (OLED) class display allowed accurate color replication. The interaction in VR requires two controllers to be held by the player. The movement of the controllers and goggles is tracked by two sensors. The game area covered about 5 m^2^, in the form of a square and determined by the location of motion sensors, as recommended by the manufacturer. The participant received visual information when approaching the boundaries of the game field. The participant’s task was to cut through blocks of various colors to the rhythm of music, which were getting closer to a participant, using two virtual swords. Blocks had to be cut with a sword whose color matched the block and in the right direction, which is indicated by an arrow on the block (Figure 1). The training session consisted of four music tracks with different pace and intensity of objects. The training session consisted of four songs: the first and second songs were “It’s time” and “Believer” performed at the normal level of difficulty. The third and fourth tracks used were “Thunder” and “Radioactive” performed at the hard level. All songs came from the band Imagine Dragons. In addition, during the song, the participant had to avoid object-obstacles, which randomly appeared in the game scenario, forcing the whole body to move.

### 2.3. Measures

The tests were performed on the first and last day. The initial tests were performed before the first training session, and the final tests were performed 30 min after the last training session.

#### 2.3.1. Reaction Time

The plate-tapping test (PTT) and the ruler-drop test (Ditrich’s test) were used to assess the reaction time.

The PTT assesses the rate of cyclic movements of the upper limb. It is based on touching two disks with maximum speed alternately, whose centers are 80 cm away from each other. The other hand is placed on a plate at equal distance from the discs during the test. The result of the test is a time of 25 touches of each disc. The test is performed by the dominant hand. This test is a part of “Eurofit Test Battery” and measures upper-body reaction time, hand–eye quickness, and coordination [22].

Ditrich’s test evaluates the reaction time and is performed using a 1.5 cm diameter and 50 cm long stick with a marked scale (every cm). The subject sits on a chair with his/her face towards the backrest, supporting the forearm of one of the upper limbs in the middle of its length. Fingers are arranged by joining the thumb and index finger in a circle, while inside, without a grip, the stick is placed. The tester holds the baton by its upper end and turns on the stopwatch to release the grip of the stick after an appropriate time. The task of the tested person is to react as quickly as possible and grasp the stick by clenching the hand. After catching, the tester reads the result on the stick. The measurement is repeated 10 times for the right hand and 10 times for the left hand. According to the methodology, out of the 10 measurements carried out, two extreme results are rejected [23].

#### 2.3.2. Hand–Eye Coordination

Trial-making test (TMT) A and TMT B were used to assess coordination and visual attention. The task of the tested in the first part (TMT A) is to combine 25 fields in order from the smallest to the largest continuous line. In the second part (TMT B), the subject has to alternate the numbers with the letters of the alphabet according to the formula 1-A, 2-B, 3-C, etc., with a continuous line. The result of the test is the time it takes for the tested person to connect all the fields in the right order [24]. TMT A is generally presumed to be a test of visual search and motor speed skills, whereas TMT B is considered to be a test of higher-level cognitive skills, such as mental flexibility.

#### 2.3.3. Energy Expenditure

The SenseWear Armband (BodyMedia, Inc, Pittsburgh, PA USA) was used to assess energy expenditure, as a secondary outcome. The device was placed on the arm of the dominant upper limb. It continuously measured physiological variables through algorithms that determined the energy expenditure expressed in kcal. The validity of the SenseWear Armband was established in a previous study [25].

### 2.4. Statistical Analysis

All statistical analyses were performed using Statistica 13.1 software (StatSoft, Cracow, Poland). The statistical significance level was set at α = 0.05. Differences between hand–eye coordination and reaction time pre-post variables were compared using Student’s paired *t*-test or the Wilcoxon signed-rank test. One-way analysis of variance was used to compare differences in the energy expenditure between training days. Continuous variables were presented as mean ± standard deviation (SD) or median and interquartile range (IQR), where appropriate, according to the Kolmogorov–Smirnov normality test. The sample size was calculated based on previous studies according to the changes in reaction time of immersive VR with an effect size of 0.795; it was determined that 12 patients should be enrolled [26]. G*Power 3.1.9 software was used to calculate the sample size. The calculation was based on Wilcoxon signed-ranks test; the type I error rate was set at 5% (α = 0.05); the effect size of the main outcomes was 0.795; the type II error rate gave 80% power.

## 3. Results

Results are presented as mean (±SD) or median and IQR. The analysis of hand–eye coordination showed significant improvement in TMT A (*p* < 0.002) and TMT B (*p* < 0.001). A significant decrease in the time to complete the task was observed.

Regarding the reaction time assessed by the PTT, a statistically significant improvement was noticed (*p* < 0.0001). The analysis of the results of Ditrich’s test for the right hand (*p* < 0.847) revealed no significant changes, while changes for the left hand showed statistical significance (*p* < 0.025) (Table 1).

Analysis of the results of energy expenditure showed that, during 5 days of training, the total average energy expenditure was 320 kcal, approximately 64 kcal per training session. We noticed a statistically significant difference between the first and the last day of training (*p* < 0.008).

## 4. Discussion

This study was aimed to investigate the potential of the use of immersive VR training for hand–eye coordination and reaction time. The results showed that the use of the VR music game selected for the study improved hand–eye coordination, as well as reaction time. It may be presumed that the improvement occurred due to an improvement in neuroplasticity. Plastic changes in the brain triggered by listening to music were evidenced by the rich history of structural and functional neuroimaging studies of the past few decades [27]. In the musicians, key hubs emerged during passive listening to music, which lie in the cerebral sensorimotor regions, whereas the dominant hubs in non-musicians lie in the parietal and left-hemispheric temporal regions [28]. Moreover, attentive music listening recruits multiple forms of working memory, attention, semantic processing, target detection and motor function, relying mainly on bilateral brain areas [29]. The additional effect obtained with this type of VR system seems to be related to the reduction in anxiety and the improvement in performance quality [30]. Training in simulated performance environments through VR systems in which conditions of “real” performance could be recreated seems to be decisive in reducing anxiety when musicians face, for example, a large audience [31]. We suppose that the cutting of blocks at different levels of difficulty (rates and intensity of their appearance) together with the forced direction of movement of the upper limbs, in particular the crossing of the body’s midline, combined with rhythmic music, could improve the interhemispheric connections. We also consider that functional reorganization may have led to structural adaptation. Thus, bimanual music game training in immersive VR may cause an increase in cortical functionality for symmetric areas involved in motor, auditory and visuo-spatial processing, as well as in the white matter tracts of the corpus callosum, similar to bimanual instrument training, such as for the piano [12]. You et al. (2005) have shown that the contact with the virtual environment provided by VR has the ability to increase activation in the primary motor cortex, which is responsible for motor performance [32]. Thus, VR interventions in our study might produce changes in cortical functional interactions by increasing the activation of the specific brain areas responsible for movement control reflected in the accelerated reaction time of participants. There is evidence of training and positive transfer from virtual to real-world environments, supporting the use of these novel virtual exercises to improve measures of sensorimotor control [33,34]. We also noticed a significant decrease in energy expenditure during the entire research week. The estimated energy expenditure was on average 64 kcal (about 4 MET) during a single practice, which corresponds to sports activities such as bicycling, brisk walking, table tennis, leisure volleyball or ballroom dancing [35]. This may have been explained by the fact that, at the beginning of the training, the VR game was a completely new experience for people participating in the study. Players familiarized themselves with the game, resulting in cutting the blocks more efficiently. This may also indicate that, during the training, participants improved hand–eye coordination and reaction time due to the visual stimulus in the VR game, thus they no longer had to put as much physical effort into the training as during the first days of the study. Interestingly, the results of reaction time evaluated with the Ditrich’s test showed a statistically significant improvement only for the non-dominant hand (as declared by the participants). Thus, it appears that, despite students using both hands to play an instrument, a preference to perform more tasks with the preferred one matters in her dexterity. Previous research indicates that left- or mix-handed musicians revealed a lesser degree of hand skill asymmetry than right-handed musicians [36]. Therefore, it can be concluded that the use of VR-music training can have a positive impact on the improvement of overall human dexterity.

Yet, these assumptions are only theoretical. An evident confirmation would certainly be to perform an fMRI, but this was not the purpose of our study. There are also no scientific reports analyzing the use of musical games for reaction time and hand–eye coordination. However, some studies suggest that these capabilities have been improved by other training sessions conducted in VR [37,38]. Previous literature revealed that action video game players showed faster response times [39], improved attention [40], faster processing speeds [41], improved multitasking abilities [42] and visuo-motor reaction time and balance performances [43]. The interest in using VR for training purposes has been described in the medical field. The results of a study by Middleton et al. (2013) showed improvement of non-dominant hand performance in a laparoscopic VR simulator after playing the Nintendo Wii [19]. Harrington et al. noticed a significant role in training medical students using laparoscopic video games (Underground) to acquire virtual laparoscopic skills [20]. Training with action video games has also yielded rehabilitation efficacy for a myriad of patients, such as stroke patients and individuals with vestibular and balance deficits [44]. Our results are in line with the results of Shin et al. (2015), who showed an improvement in hand–eye coordination in 16 children with cerebral palsy using Nintendo Wii VR equipment. In contrast to our study, the trainings were held twice a week for 45 min each session and lasted 8 weeks [45]. Pourazar et al. (2018) likewise examined the impact of VR intervention training on reaction times in children with cerebral palsy. The results revealed improvement in the experimental group after the VR intervention program. The authors suggested VR as a promising tool in the rehabilitation process to improve the response time in children with cerebral palsy [26]. Similar findings can be seen in the research of Straker et al. (2011), who reported that VR devices may have a positive impact on the physical and mental health, especially visual and spatial integration, in children with developmental coordination disorders [46]. We also found two studies using the Beat Saber game for upper limb training after stroke [47] and the consequences of playing exergames on aspects of vision, cognition, and self-reported VR sickness [48]. Erhardsson et al. (2020) stated that commercial head-mounted-display (HMD) VR might be a useful tool for people with chronic stroke to improve their upper extremity functioning. Szpak et al. (2020) did not observe evidence of adverse events 40 min after exit from VR, irrespective of the duration of the intervention (10 or 50 min).

In accordance with the hypothesis, the VR training improved the hand–eye coordination and reaction time of musicians, which may lead to the faster mastering of a musical instrument. Throughout the study, the participants were enthusiastic to participate, day by day more involved in the game, and enjoyed playing according to their subjective opinions. In contrast, control tests were rated as boring. Participants did not indicate any complaints about the handling of the equipment, nor the performance of the game tasks. No adverse effects after training were reported. Unfortunately, we did not use standardized scales, only the subjective feelings of the participants. Furthermore, participants indicated that they would be happy to participate again but also that the training program could last longer. We believe that the practical aspect of our research may be of great importance in the education of young musicians, as it was shown that musicians who start training at an early age exhibit greater centrality in the auditory cortex, as well as areas related to top-down processes, attention, emotion, somatosensory processing and non-verbal processing of speech. Considering that similar effects on the human brain have an interaction in a VR environment, it can be assumed that traditional education in a music school together with entertainment in the form of immersion music game may stimulate the brain’s neuroplasticity to an even greater degree.

Although this study provides encouraging results, we recognize that some limitations should be considered. Firstly, this study was designed as a single-group pre-post study without the inclusion of a control group. Secondly, the research included a small study group. Third, more standardized research methods should be used in future studies for the assessment of handedness (i.e., the Edinburgh handedness inventory test) and cyber-sickness (i.e., Virtual Reality Sickness Questionnaire). Finally, the follow-up assessment could provide additional valuable information on the effectiveness of VR systems, especially when the system applies music in an immersive way.

## 5. Conclusions

The results showed that off-the-shelf, room-scale head-mounted-display VR has the potential to improve hand–eye coordination and reaction time in young musicians. The section of available games must be appropriate, and free software will allow the adaptation of different environments and types of music to the needs of each individual. The promising results shown may guide the future start-up of randomized controlled studies that can show its efficacy in this population.

## Figures and Tables

**Figure 1 ijerph-18-01297-f001:**
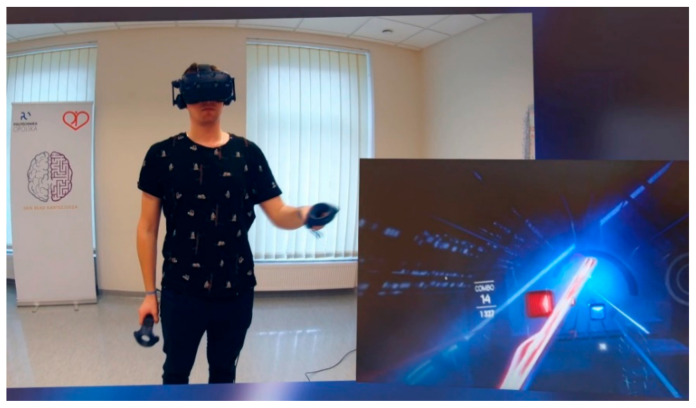
Example of a training session.

**Table 1 ijerph-18-01297-t001:** Results of the study.

Variable	Test	Value	*p*
Hand–eye coordination (s)	TMT A pre-	19.2 ± 7.2	0.002 ^a^
	TMT A post-	14.4 ± 3.9	
	TMT B pre-	29.4 (23.2–32.1)	0.001 ^b^
	TMT B post-	19.7 (18.5–26.4)	
Reaction time (s)	PTT pre-	12.6 ± 1.7	0.0001 ^a^
	PTT post-	10.8 ± 1.2	
Reaction time (cm)	Ditrich’s right hand pre-	17.0 ± 5.9	0.847 ^a^
	Ditrich’s right hand post-	16.9 ± 5.4	
	Ditrich’s left hand pre-	16.6 ± 5.3	0.025 ^a^
	Ditrich’s left hand post-	15.6 ± 5.7	
Energy expenditure (kcal)	Day 1 *	80.2 ± 28.7	0.011 ^c^
	Day 2	68.6 ± 20.6	
	Day 3	62.6 ± 18.6	
	Day 4	57.6 ± 18.9	
	Day 5 *	51.2 ± 21.6	

^a^ Student’s *t*-test; ^b^ Wilcoxon test; ^c^ one-way analysis of variance; * *p* < 0.05.

## Data Availability

The data presented in this study are available on request from the corresponding author.

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
