# Peer review of "Training Using a Commercial Immersive Virtual Reality System on Hand–Eye Coordination and Reaction Time in Young Musicians: A Pilot Study"

_ijerph, 2021, doi:10.3390/ijerph18031297_

Round 1

Reviewer 1 Report

Dear Authors,

I carefully reviewed your manuscript. I believe the topic of your study is original. The article is generally well written.

I reject your article for publication on this journal.

Unfortunately, the study presents some methodological limitations, such approach limits the validity of the discussion of the results achieved. The main methodological issues are the following: the low number of subjects, the lack of power analysis. In addition, the authors presented results at short-term follow-up and it could be interested to evaluate a   further follow-up to evaluate during the time the improvements of the subject. Moreover, this study would have greater relevance comparing with a control group.

I suggest you increasing the number of the subject and providing more demographic (age and gender) or participant profile details.

Author Response

Dear Reviewer, thank you for your comment. Authors agree with the reviewer that the paper has the weakness of a small sample size and that a comparison of results with a control and follow-up would provide valuable information. However, we are unable to perform such a study at this time. The pandemic has shifted music school teaching remotely. We plan to conduct a randomized controlled trial in the future, but in our opinion it seems reasonable to publish the results of this pilot study. The development of VR is very dynamic, but no one in the available literature has conducted a similar study before. Promising results may make VR teams pay attention to this aspect. Another important effect of the publication of this study may be the start of training in the home environment of young musicians. We assume that most of them have access to a console/computer where they could play such a musical game - which would foster (in our opinion) their ability to play an instrument. Moreover, as we have shown the estimated energy expenditure was on average 64 kcal (about 4 MET) which corresponds to sports activities like bicycling, brisk walking, table tennis, leisure volleyball or ballroom dancing. In the face of widely reported reduced physical activity level during eLearning, this seems significant.

The paper has been extensively evaluated and many changes have been made, which seem to meet the conditions of the pilot/feasibility study.

Reviewer 2 Report

The paper present a pre-post test related to the use of an immersive VR music game with the aim evaluating the improvement of hand-eye coordination and dexterity.

The study is limited in scope but it is well justified, adequately conducted, well presented, and discussed with extensive references to related literature. I believe the contribution may be considered significant not only by people interested in music teaching/learning but also - more in general - by people interested in hand-eye coordination and dexterity issues.

Minor comments:

  • in the discussion authors write: "The results showed that the use of a VR music game improve hand-eye coordination, as well as reaction time."  I think that the statement should be lessened to something like: "The results showed that the use of the VR music game selected for the study improve hand-eye coordination, as well as reaction time." 
  • the Conclusion section is a bit too concise 

Author Response

Dear Reviewer, thank you for your comment. Below are our responses to remarks on a point-by-point basis.

"in the discussion authors write: "The results showed that the use of a VR music game improve hand-eye coordination, as well as reaction time."  I think that the statement should be lessened to something like: "The results showed that the use of the VR music game selected for the study improve hand-eye coordination, as well as reaction time." 

Thank you for that suggestion. We agree that the sentence was too general, we have completed it as indicated.

Reviewer 3 Report

Line 54: Add a reference.

Lines 68-70: please cite the following article when referring to immersion and VR: Slater, Immersion, and the illusion of presence in virtual reality, Br. J. Psychol. 109 (2018) 431–433. https://doi.org/10.1111/bjop.12305.

Lines 70-72: Please add these studies regarding immersive VR and motor rehabilitation:

Matamala-Gomez, M., De Icco, R., Avenali, M., & Balsamo, F. (2018). Multisensory integration techniques in neurorehabilitation: The use of virtual reality as a rehabilitation tool.

Matamala-Gomez, M., Malighetti, C., Cipresso, P., Pedroli, E., Realdon, O., Mantovani, F., & Riva, G. (2020). Changing body representation through full body ownership illusions might foster motor rehabilitation outcome in patients with stroke. Frontiers in Psychology11.

Further, I suggest to the authors to change their definition of immersion in VR following the definition of this manuscript.

Lines 75-77: Is this statement a personal opinion or it is based on some scientific work? If yes cite the manuscript at the end of the statement. If it is not, then I suggest the authors starting the sentence like this: “We hypothesize….”

Lines 78-80: The authors should cite the studies that they are referring to…

Lines 80-81: Again, which are the studies that you found? Cite them.

Lines 192-194: Please justify that statement by citing some studies reporting this finding.

Line 207: You et al. (add here the year of the study).

Line 220: The author should also discuss why participants improved left-hand dexterity at the end of the training but not right-hand dexterity. Did the authors control participants' handedness with some screening tests like the Edinburgh handedness inventory test? If not the author should add this fact as a limitation of the study and hypostasize that founded differences between right and left-hand dexterity improvement may be due to the participants' handedness.

Line 230: Middleton et al. (add here the year of the study). Repeat this throughout the whole manuscript.

Author Response

Dear Reviewer, thank you for your comment. Below are our responses to remarks on a point-by-point basis.

  1. Line 54: Add a reference.

Done

  1. Lines 68-70: please cite the following article when referring to immersion and VR: Slater, Immersion, and the illusion of presence in virtual reality, Br. J. Psychol. 109 (2018) 431–433. https://doi.org/10.1111/bjop.12305.

Done

  1. Lines 70-72: Please add these studies regarding immersive VR and motor rehabilitation

Done. We supplemented this in the discussion in the sections on motor cortex stimulation (page 6, lines 223-225).

  1. Lines 75-77: Is this statement a personal opinion or it is based on some scientific work? If yes cite the manuscript at the end of the statement. If it is not, then I suggest the authors starting the sentence like this: “We hypothesize….”

Thank you for this suggestion. This is our hypothesis because there is no work linking VR and mastering a musical instrument. We have adjusted the sentence.

  1. Lines 78-80: The authors should cite the studies that they are referring to…

Thank you for pointing this out, done.

  1. Lines 80-81: Again, which are the studies that you found? Cite them.

Thank you for this suggestion, however this sentence does not refer to any papers - it indicates a lack of papers. The previous sentence indicates (81-83) that coordination in medics has been evaluated, while the next sentence indicates that there is no research in other professions where hand-eye coordination is relevant (line 83-84). Thus, we left the sentence as it is.

  1. Lines 192-194: Please justify that statement by citing some studies reporting this finding.

Thank you for pointing this out, done.

  1. and 10. Line 207: You et al. (add here the year of the stud); Line 230: Middleton et al. (add here the year of the study). Repeat this throughout the whole manuscript.

Done

  1. Line 220: The author should also discuss why participants improved left-hand dexterity at the end of the training but not right-hand dexterity. Did the authors control participants' handedness with some screening tests like the Edinburgh handedness inventory test? If not the author should add this fact as a limitation of the study and hypostasize that founded differences between right and left-hand dexterity improvement may be due to the participants' handedness.

Thank you for pointing this out. As we indicated in the manuscript the left hand was non-dominant, based on the declarations that participants made. All declared the right hand as dominant. Thank you for pointing out the tool, we will certainly use this in the continuation of the study. We have supplemented the discussion (lines 235-240) and limitation (lines 294-296) with this aspect.

Round 2

Reviewer 1 Report

Dear Authors,

I carefully reviewed your manuscript. I believe the topic of your study is original. The article is generally well written.

The paper has consistently improved in quality and meet the conditions of the pilot study

I approve the publication of this paper with minor revision.

Abstract

L.21 The word “showed” there is twice

L.22 “only several”, please use another term (for example. Limited or small number of …)

Introduction

L.78-80 please move the Hypothesize and put together with the other one in line 90-92

Methods.

L.96 Please, check the age “…. 14 to 19 years (8 females, 6 males, age = 22.2±1.3 years). …. “The inclusion criteria included individuals aged 14 to 20 years”

L.168 “The”. I suppose you mean “that”

Discussion.

L.229 “environ-ments”. I suppose you mean “environments”

L.251 with “some studies” you should cite at least 2 studies

Was the training performed every session at the same time for all the subjects?

Conclusion

L.309 Please rephrase with “These results may…” or “The results showed”

Author Response

We appreciate your valuable comments. Your suggestions are very practical. We are sorry for these spelling and editorial errors. We meticulously revised them with the text highlight colour of yellow